# Acute Kidney Injury in Children with Acute Appendicitis

**DOI:** 10.3390/children9050620

**Published:** 2022-04-27

**Authors:** Pierluigi Marzuillo, Crescenzo Coppola, Roberta Caiazzo, Giulia Macchini, Anna Di Sessa, Stefano Guarino, Francesco Esposito, Emanuele Miraglia del Giudice, Vincenzo Tipo

**Affiliations:** 1Department of Woman, Child and of General and Specialized Surgery, Università Degli Studi Della Campania “Luigi Vanvitelli”, Via Luigi De Crecchio 2, 80138 Naples, Italy; anna.disessa@unicampania.it (A.D.S.); stefano.guarino@policliniconapoli.it (S.G.); emanuele.miraglia@unicampania.it (E.M.d.G.); 2Pediatric Emergency Department, A.O.R.N Santobono-Pausilipon, Via Mario Fiore 6, 80129 Naples, Italy; renzocoppola90@gmail.com (C.C.); robertacaiazzo988@gmail.com (R.C.); giulia.macchini@gmail.com (G.M.); enzotipo@libero.it (V.T.); 3Pediatric Radiology Unit, A.O.R.N Santobono-Pausilipon, Via Mario Fiore 6, 80129 Naples, Italy; frain.rx@gmail.com

**Keywords:** acute kidney injury, acute appendicitis, children

## Abstract

We hypothesized that—as in other common pediatric conditions—acute appendicitis (AA) could be complicated by acute kidney injury (AKI). We aimed to investigate the prevalence of, and the factors associated with AKI in a cohort of patients with AA. We retrospectively collected data of 122 children (63.9% of male gender; mean age 8.6 ± 2.9 years; range: 2.2–13.9 years) hospitalized for AA. AKI was defined according to the Kidney Disease/Improving Global Outcomes creatinine criteria. We considered a basal serum creatinine value as the value of creatinine estimated with the Hoste (age) equation, assuming that the basal estimated glomerular filtration rate (eGFR) was 120 mL/min/1.73 m^2^. Explorative univariate logistic regression analysis was used to explore the associations with AKI. Out of 122 patients, nine (7.4%) presented with AKI. One patient had stage two AKI and the remaining had stage one AKI. The maximum AKI stage was found at admission. The patients with AKI showed a higher prevalence of fever ≥ 38.5 °C (*p* = 0.02), vomiting (*p* = 0.03), ≥5% dehydration (*p* = 0.03), and higher levels of both C-reactive protein (CRP) (*p* = 0.002) and neutrophils (*p* = 0.03) compared with patients without AKI. Because all patients with AKI also presented with vomiting, an Odds Ratio (OR) for the vomiting was not calculable. The exploratory univariate logistic regression analysis confirmed that fever ≥ 38.5 °C (OR = 5.0; 95% CI: 1.2/21.5; *p* = 0.03), ≥5% dehydration (OR = 8.4; 95% CI: 1.1/69.6; *p* = 0.04), CRP (OR = 1.1; 95% CI: 1.05/1.2; *p* = 0.01), and neutrophil levels (OR = 1.1; 95% CI: 1.01/1.3; *p* = 0.04) were all predictive factors of AKI. AKI can occur in 7.4% of patients with AA. Particular attention should be paid to the kidney health of patients with AA especially in the presence of vomiting, ≥5% dehydration, fever ≥ 38.5 °C, and high CRP and neutrophils levels.

## 1. Introduction

Acute appendicitis (AA) is the most common surgical pediatric emergency with about 250,000 cases/year in the United States [1].

The management of AA could be conservative with broad spectrum antibiotics administrations if the inflamed appendix is intact (uncomplicated) or surgical if the inflamed appendix has developed perforation and/or gangrene or has developed into an appendiceal mass or abscess (complicated) [2]. When the diagnosis is performed, perforation could be already present in 30–75% of children, with young children being at a higher risk [3]. Perforated appendicitis increases the morbidity with an intra-abdominal abscess being an important complication [3].

Acute kidney injury (AKI) is a common complication in several pediatric diseases. In fact, AKI can complicate the acute gastroenteritis (AGE) in about 25% of the cases [4], the community acquired pneumonia (CAP) in about 20% of the cases [5], and the type 1 diabetes mellitus (T1DM) onset in about 44% of the cases, increasing up to 65% when T1DM onset is characterized by diabetic ketoacidosis [6,7].

We hypothesized that —as in other common pediatric conditions [4,5,6]—AA could also be complicated by AKI with a similar prevalence to that found in AGE [4] and CAP [5] in childhood. Moreover, because AA can manifest with vomiting and/or diarrhea and with important systemic inflammation, we hypothesized that the development of AKI in children with AA could be linked both to dehydration (as occurs in case of AGE) and to inflammation (as occurs in case of CAP).

Because no similar studies are available, we aimed at investigating the prevalence of, and the factors associated with AKI in a cohort of patients with AA.

To not miss an AKI diagnosis is relevant because even a mild AKI episode doubles the risk of chronic kidney disease [8]. To identify patients with AA having presented an AKI could be useful to plan a proper nephrological follow-up. A specific follow-up, in fact, has been associated with improved outcomes after severe AKI in adults [9].

## 2. Methods

We retrospectively collected the data of all patients aged < 14 years consecutively admitted for AA to the Pediatric Emergency Department of AORN Santobono-Pausilipon, Naples, Italy, from 1 June 2020 to 31 December 2020. Our Research Ethical Committee approved the study (approval n°12770/2020), and informed consent was obtained before any procedure.

Inclusion criteria were (i) discharge primary diagnosis of AA associated to surgical confirmation or—for the conservatively managed patients—to ultrasound evidence of non-compressible appendix > 6 mm in diameter, and (ii) availability of creatinine serum levels at admission. We excluded 15 patients with concomitant COVID-19 infection because this virus has been implicated in AKI development [10]. We also excluded 20 patients with missing data about serum creatinine values at admission. None of the patients had previously known nephro-urological diseases. We recorded all the serum creatinine values obtained during hospitalization. Because AKI has been described in 3.2% of children undergoing noncardiac surgery [11], we evaluated other creatinine values only if measured before surgery in order to be sure to detect AKI episodes only related to AA and not to the surgical procedures.

### 2.1. Data Collection

From the digitalized clinical charts, we collected the clinical and laboratory characteristics listed in Table 1, as well as the history of known nephro-urological disease.

### 2.2. Definitions

Serum creatinine concentration was measured by an isotope-dilution mass spectrometry (IDMS)-traceable method [12].

AKI was defined only by the Kidney Disease/Improving Global Outcomes (KDIGO) serum creatinine criteria because the urinary output measurement was lacking [13]. The height was missing in part of clinical charts. Therefore, considering that height-dependent and height-independent basal serum creatinine estimation methods are comparable [14], we back-calculated the basal serum creatinine by the Hoste (age) equation [12]. As none of the enrolled children was aged < 2 years, we assumed as basal estimated glomerular filtration rate (eGFR) an eGFR of 120 mL/min/1.73 m^2^ [14].

No AKI, stage 1, stage 2, or stage 3 AKI were defined by serum creatinine values < 1.5, 1.5 to <2, 2 to <3, and ≥3 times the basal serum creatinine, respectively [13].

The dehydration was classified in <5%, or ≥5% of fluid deficits on the basis of retrospective evaluation of clinical conditions reported in clinical charts and according to the World Health Organization definition [15]. If the patient has 2 signs among irritability, sunken eyes, thirstiness, and skin turgor that goes back slowly, they are at least moderately dehydrated (≥5% of fluid loss).

### 2.3. Post-Hoc Power Calculation

No studies investigated the AKI prevalence in adults/children with AA. Studies in children with pneumonia and acute gastroenteritis reported an AKI prevalence ranging from 20.4 to 24.6% [4,5]. On the basis of the mean AKI prevalence of 22.5% in these pediatric populations [4,5], considering the prevalence of AKI of 7.4% in our population of 122 subjects, the calculated Post-Hoc power, with an alpha of 0.05, was >90%.

### 2.4. Statistical Analysis

*p* values < 0.05 were considered significant. Differences for continuous variables were analysed with the independent-sample *t*-test for normally distributed variables and with the Mann–Whitney test in case of non-normality. Qualitative variables were compared using the Chi–squared test or Fisher exact test where applicable. Logistic regression was used with the aim of exploring associations with AKI. We added in the explorative univariate logistic regression analysis the parameters that associated (*p* < 0.05) to AKI after comparison of characteristics of the patients with and without AKI.

## 3. Results

One-hundred and fifty-seven patients were eligible. After the exclusion of 15 patients with concomitant COVID-19 infection and of 20 patients with missing data about serum creatinine values at admission, we enrolled 122 patients (78 males) with AA diagnosis and a mean age of 8.6 ± 2.9 years (age range: 2.2–13.9 years) (Table 1).

At admission, all the patients underwent intravenous hydration with Glucose 5% Sodium Chloride 0.9%, independently from the chosen treatment modality. The duration of intravenous fluids was of 3.7 days ± 0.7 standard deviation scores (SDS) (Table 1).

All the enrolled patients underwent abdomen ultrasounds and in none of them anomalies of the kidney or of the urinary tract were found.

As per the inclusion criteria, in all the 84 patients undergoing surgical intervention the AA was pathologically confirmed (39 had complicated—by perforation—AA and 41 had uncomplicated AA) and, in the other 38 conservatively managed patients, non-compressible appendix > 6 mm in diameter was found at abdomen ultrasound.

Five out of 38 (13.2%) conservatively managed patients presented with AKI compared with four out of 84 (4.8%) patients undergoing surgical treatment (*p* = 0.13). Moreover, one out of 39 (2.6%) patients with complicated AA presented with AKI compared with eight out of 83 (9.6%) patients with uncomplicated AA (*p* = 0.27).

As per our clinical practice, the patients were conservatively managed in case the clinical findings were suggestive of non-complicated AA (appendicitis without either perforation nor appendiceal abscess nor mass formation) [16]. Because often the differentiation between uncomplicated and complicated or perforated appendicitis may be uncertain before the operation [16], in case of uncertainty, the surgical treatments were chosen.

All the patients undergoing conservative management were treated with intravenous cephalosporins and metronidazole for 3–5 days followed by oral amoxicillin and clavulanic acid for 10 days. 

Out of the 122 patients, nine (7.4%) presented with AKI, one with stage 2, and the remaining with stage 1 AKI. All patients showed the maximum AKI stage at admission.

Patients with AKI showed a higher prevalence of fever with axillar body temperature ≥ 38.5 °C, vomiting, and ≥5% dehydration, and higher levels of both C-reactive protein (CRP) and neutrophils compared with patients without AKI (Table 1).

Because all patients with AKI also presented with vomiting, an Odds Ratio for the vomiting was not calculable (Table 2). The exploratory univariate logistic regression analysis confirmed that a body temperature ≥ 38.5 °C, ≥5% dehydration, CRP, and neutrophils were all predictive factors of AKI (Table 2).

Evaluating our hospital registry, none of the patients having developed AKI when returned at specific nephrological follow-up visits or underwent visits for having developed hypertension, proteinuria, or reduced eGFR during a post-discharge period of 12 months.

## 4. Discussion

This pilot retrospective study is the first report investigating the prevalence of AKI in patients with AA. Only few case reports describing AKI following bilateral ureteral obstruction as a post-appendicectomy complication in childhood are available [17]. We found that about 7% of children with AA could present with AKI and then with increased serum creatinine values at admission. This prevalence is much more than the prevalence of chronic kidney disease in childhood which is of 74.7 per million of children in our country [18]. This reinforces a causal relationship between AA and AKI.

Before performing this study, we expected a higher prevalence of AKI in this category of patients with a hypothesized prevalence of 20–25%, similarly to the findings in patients with CAP and AGE [4,5]. We hypothesize that the lower AKI prevalence compared with these conditions could be linked to the usual prompt explosion of AA symptoms determining a quick diagnosis and a prompt treatment. In fact, patients with AA were hospitalized after only a median of 20 h with a prompt start of intravenous fluid administration compared with a median duration of symptoms of about 48 h and 120 h, respectively, for AGE [4] and CAP [5]. The fact that patients hospitalized for AA presenting with AKI showed a trend indicating a longer median duration of symptoms (30 h vs. 20 h) compared with those without AKI, partially further confirm this hypothesis (Table 1). Moreover, the only patient with stage two AKI had a symptoms duration of 72 h while patients with stage one AKI had a symptoms duration ranging 6–48 h. Similarly, in other pediatric conditions, patients with AKI showed a longer symptoms duration compared with those without AKI [4,5,6]. Interestingly, among patients with AA, AKI was associated with the presence of vomiting, ≥5% dehydration, an axillary body temperature ≥ 38.5 °C, and with higher levels of CRP and neutrophils. This could indicate that in the AKI pathophysiology in the setting of AA, it could be relevant that there is both a “pre-renal” component related to dehydration as in the case of AGE [4] and T1DM [6], and a systemic inflammatory response as in the case of CAP [5].

This could imply that a precocious rehydration could be important in the patients with AA presenting with vomiting and/or with ≥5% dehydration, in order to prevent as best as possible a concomitant AKI episode. Moreover, a proper and prompt AA treatment aiming to reduce the systemic inflammation could also be useful in the AKI prevention.

Evaluating the duration of intravenous fluids administration, no difference was found comparing patients with and without AKI (Table 1). This could be related to the fact that the AKI developed before hospitalization and to the fact that, after the admission, the standardized rehydration protocol applied in our hospital and the prompt AA treatment prevented new onsets of AKI.

We found a trend indicating a higher prevalence of AKI among conservatively managed patients compared with those undergoing surgical intervention. This could be justified by the slower resolution of inflammation and the disappearance of AA symptoms among conservatively managed patients. In fact, evidence indicates that conservatively treated patients with AA experience an increased length of hospital stay with the length depending on the regimen and type of antibiotic prescribed [16].

Similarly, we also found a trend indicating the lower prevalence of AKI among children with (usually surgically treated), compared with those without complicated appendicitis (usually conservatively treated).

Moreover, as per our clinical practice, none of the retrospectively enrolled patients were treated with nephrotoxic antibiotics, and as such their role in AKI development can be excluded among enrolled patients. However, the protocol of AA conservative management varies among groups and also the utilization of nephrotoxic antibiotics such as gentamicin have been considered [16]. It could be interesting to clarify the role of nephrotoxic antibiotics in future research.

Finally, we want underline that the AKI diagnoses were made only a posteriori when we decided to make this study and in none of them we diagnosed AKI during hospitalization. To underline this concept is important because in several pediatric conditions AKI can develop but often it is not suspected and then not recognized [5].

To not miss an AKI episode, however, is relevant, because to have presented with a mild AKI episode doubles the risk of chronic kidney disease during the follow-up [8] and a specific post-discharge nephrological follow-up for patients having presented with AKI improves the long-term outcomes [9].

The main limitation of this pilot study is the lack of a measured basal serum creatinine (before hospitalization). To have a reliable basal creatinine, however, the biochemical samples should have been collected near the manifestation of AA and unfortunately this is impossible, also if a prospective study will be designed because we cannot predict when an acute appendicitis will manifest itself. Because the basal serum creatinine is often unknown in childhood, for the retrospective studies in the field of pediatric AKI, methods to reliably estimate a basal creatinine have been developed. In the study by Hessey et al., the agreement between AKI defined by estimated versus measured basal serum creatinine was high (>80% (κ > 0.5)) [14]. Moreover, the estimated basal creatinine to diagnose AKI when basal serum creatinine is unknown was used also in several retrospective studies in children [4,5,7]. Other limitations are represented by the retrospective design with the lack of a precise quantification of dehydration degrees and vomiting episodes; the single center experience; the lack of serum acid uric level measurements; the limited number of patients having developed AKI, which allowed us only to run exploratory univariate analyses; the impossibility to use the KDIGO urinary output criteria, which could have determined underestimation of AKI prevalence; the lack of a post-discharge follow-up allowing us to evaluate the percentage and the timing of AKI resolution.

This study, however, represents only a pilot observation, opening the way to prospective studies with a specific post-discharge nephrological follow-up in order to precisely evaluate the recovery from AKI and the onset of possible AKI complications, such as reduced eGFR, proteinuria, or hypertension during the follow-up.

In conclusion, we found that 7.4% of patients hospitalized with AA may suffer with mostly mild AKI. Factors associated to the AKI development were the presence of vomiting, ≥5% dehydration, an axillary body temperature ≥ 38.5 °C, and higher levels of CRP and neutrophils.

Because AKI could manifest in several common pediatric conditions [4,5,6], and because even repeated mild AKI episodes determine an increased risk of developing chronic kidney disease later in life [19], the pediatricians must be aware of this risk and make all the efforts to prevent as best as possible AKI episodes in childhood.

## Figures and Tables

**Table 1 children-09-00620-t001:** Clinical and laboratory characteristics of children hospitalized for acute appendicitis with and without AKI.

	All Patients No. = 122	AKI (no) No. = 113	AKI (yes) No. = 9	*p*
Age, yr, mean (SDS)	8.6 (2.9)	8.7 (2.8)	7.6 (3.9)	0.27
Male gender, No. (%)	78 (63.9)	72 (63.7)	6 (66.7)	0.85
T max, °C, mean (SDS)	37.4 (1.2)	37.2 (2.1)	38.5 (1.1)	0.15
T max ≥ 38.5 °C, No. (%)	38 (31.1)	32 (28.3)	6 (66.7)	0.02
Symptoms duration, h, median (IQR)	20.0 (35.0)	20.0 (35.0)	30.0 (18.0)	0.69
Vomiting, No. (%)	82 (67.2)	73 (64.6)	9 (100)	0.03
≥5% dehydration	63 (51.6)	55 (48.7)	8 (88.9)	0.03
Duration of intravenous fluids administration, days, mean (SDS)	3.7 (0.7)	3.6 (0.7)	3.8 (0.7)	0.73
Surgical treatment, No. (%)	84 (68.8)	78 (69.0)	6 (66.7)	0.88
C-reactive protein, mg/L, median (IQR) *	25.5 (94.9)	21.9 (82.7)	121.9 (56.99)	0.002
Neutrophils, n/mcL, mean (SDS) **	1246 (5296)	12,061 (5341)	15,927 (3033)	0.03
Platelets, median (IQR)	286,500 (99,000)	288,000 (99,000)	274,000 (48,000)	0.66
Glucose, median (IQR)	95.5 (21.0)	95.0 (23.0)	100.0 (7.0)	0.22
Urea, median (IQR)	24.0 (9.0)	23 (6.0)	24 (9.0)	0.58
Creatinine, mean (SDS)	0.44 (0.1)	0.43 (0.1)	0.61 (0.1)	<0.001
Na, mean (SDS)	136.4 (3.0)	136.4 (2.9)	135.9 (3.7)	0.59
K, mean (SDS)	4.0 (0.35)	4.0 (0.3)	3.6 (0.4)	0.17
Cl, mean (SDS)	98.7 (3.3)	100.0 (3.3)	98 (3.5)	0.79

* For each increase of 10 mg/L. ** For each increase of 1000/mcL. Abbreviations: AKI, acute kidney injury; IQR, interquartile range; SDS, standard deviation score; and T, body temperature.

**Table 2 children-09-00620-t002:** Exploratory univariate logistic regression analysis exploring factors associated with AKI in children hospitalized for acute appendicitis.

	OR (95%CI); *p*	95%CI	*p*
T max ≥38.5 °C, No. (%)	5.0	1.2/21.5	0.03
Vomiting, No. (%)	Not calculable ***	Not calculable ***	Not calculable ***
≥5% dehydration	8.4	1.1/69.6	0.04
C-reactive protein, mg/L, median (IQR) *	1.1	1.05/1.2	0.01
Neutrophils, n/mcL, mean (SDS) **	1.1	1.01/1.3	0.04

* For each increase of 10 mg/L. ** For each increase of 1000/mcL. *** Not calculable because all the patients with AKI presented vomiting. Abbreviations: AKI, acute kidney injury; CI, confidence interval; IQR, interquartile range; OR, odds ratio; SDS, standard deviation score; and T, body temperature.

## Data Availability

Data supporting reported results can be obtained on request.

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
