# Peer review of "Acute Kidney Injury in Children with Acute Appendicitis"

_children, 2022, doi:10.3390/children9050620_

Round 1

Reviewer 1 Report

The report was designed to evaluate the incidence of AKI in children suffering from acute appendicitis (AA). The criteria used for diagnosis of AKI come from KIDIGO guidelines, which use only a D serum creatinine concentration as a major marker, while more extended pRIFLE criteria use also urine output.

The general message of this report is that about 7% of children with AA present AKI and all of them were vomiting due to acute inflammation and were highly dehydrated because of this and due to high fever. One can presume, that pre-renal AKI was a major factor and high inflammation markers were superimposed.

The major problem with reliability of the data is that baseline creatinine concentration was not measured, but  extrapolated, therefore all consecutive comparisons of the creatinine values  to the baseline (a major KIDIGO criterion) were projected basing of “considered” values. This may create a bias to overestimate the incidence of AKI.

Missing data:

  • What a difference (if any) of AKI incidence between patients operated vs treated with antibiotics (conservatively)?
  • What antibiotics have been used ( in dehydrated children) and did (at least some of them) has a nephrotoxic potential?
  • What was a duration of time of inadequate (or not given) hydration (as all AKI patients were dehydrated)? Shortly – how long was a dehydration before introducing appropriate fluid treatment at hospital?
  • Was the diagnosis of AA confirmed in 100% of cases by operating surgeons?
  • Was the concentration of uric acid measured (one of the important factors in “surgical” AKI cases)?
  • Please define a final renal outcome in children diagnosed as AKI

Author Response

Reviewer 1

The report was designed to evaluate the incidence of AKI in children suffering from acute appendicitis (AA). The criteria used for diagnosis of AKI come from KIDIGO guidelines, which use only a delta serum creatinine concentration as a major marker, while more extended pRIFLE criteria use also urine output.

The general message of this report is that about 7% of children with AA present AKI and all of them were vomiting due to acute inflammation and were highly dehydrated because of this and due to high fever. One can presume, that pre-renal AKI was a major factor and high inflammation markers were superimposed.

The major problem with reliability of the data is that baseline creatinine concentration was not measured, but extrapolated, therefore all consecutive comparisons of the creatinine values to the baseline (a major KIDIGO criterion) were projected basing of “considered” values. This may create a bias to overestimate the incidence of AKI.

Answer: thank you for your comment. This comment gives the possibility to explain better the methodology of our study. Often, in children the basal creatinine is unknown because only in case of chronic diseases children routinely undergo biochemical exams. Moreover, to have a reliable basal creatinine, the biochemical samples should have been collected near the manifestation of acute appendicitis and unfortunately this is impossible because we cannot predict when an acute appendicitis will onset. For this reason, for the retrospective studies in the field of pediatric AKI, methods to reliably estimate a basal creatinine have been developed. In the study by Hessey et al. (Pediatr. Nephrol. 2017, 32, 1953–1962, doi:10.1007/s00467-017-3670-z.) the agreement between AKI defined by estimated versus measured basal serum creatinine was high (>80% [κ >0.5]). Moreover, we used this method to diagnose AKI when basal serum creatinine was unknown also in other retrospective studies about AKI in children (Pediatr. Nephrol. 2021, 36, 2883–2890, doi:10.1007/s00467-021-05022-x; Pediatr. Nephrol. 2021, 36, 1627–1635, doi:10.1007/s00467-020-04834-7.). We added these considerations among limitations of the study, please see lines 232-242 of the new version of the manuscript).

Missing data:

What a difference (if any) of AKI incidence between patients operated vs treated with antibiotics (conservatively)?

Answer: we added this information and we discussed it in the new version of the manuscript (please see lines 126-134 and 207-215).

What antibiotics have been used (in dehydrated children) and did (at least some of them) has a nephrotoxic potential?

Answer: as per our clinical practice all the patients undergoing to conservative management were treated with intravenous cephalosporins and metronidazole. Therefore, none of them was treated with nephrotoxic antibiotics. We added this information in the new version of the manuscript (please see lines 135-137 and 217-222 of the new version of the manuscript)

What was a duration of time of inadequate (or not given) hydration (as all AKI patients were dehydrated)? Shortly – how long was a dehydration before introducing appropriate fluid treatment at hospital?

Answer: At admission, independently from the chosen treatment modality, all the patients underwent intravenous hydration with Glucose 5% Sodium Chloride 0.9%. For this reason, the duration of symptoms (and then of dehydration) is that shown in the Table 1 in the row “Symptoms duration, h, median (IQR)”. We added this information in the new version of the manuscript (please see lines 117-119 and 182-183 of the new version of the manuscript). We also added information about the duration of intravenous liquids administration and we discussed it in the discussion section (please see lines 118-119 and 202-206 of the new version of the manuscript).

Was the diagnosis of AA confirmed in 100% of cases by operating surgeons?

Answer: One of the inclusion criteria was a discharge primary diagnosis of acute appendicitis associated to surgical confirmation. For this reason, all the enrolled patients undergoing surgical intervention presented confirmation of acute appendicitis. This information is available at lines 67-68 and 122-125.

Was the concentration of uric acid measured (one of the important factors in “surgical” AKI cases)?

Answer: the uric acid measurement is not available because it is not routinely dosed in our clinical practice for the patients with the suspect of acute appendicitis. We added this among limitations of the study (please see lines 244-245 of the new version of the manuscript).

Please define a final renal outcome in children diagnosed as AKI

Answer: as stated among limitation of the study, we lack of a post-discharge follow-up allowing us to evaluate the percentage and the timing of AKI resolution (please see lines 248-249). However, analyzing our hospital registry, none of the patients with the AKI diagnosis following AA returned to pediatric nephrological follow-up for reduced estimated glomerular filtration rate, proteinuria or hypertension. We added this information on the new version of the manuscript. Please see lines 165-167 of the new version of the manuscript. We also underlined that this study represents a pilot observation opening the way to prospective studies with a specific post-discharge nephrological follow-up in order to precisely evaluate the recovery from AKI and the onset of possible AKI complications such as reduced eGFR, proteinuria or hypertension during the follow-up (please see lines 251-254 of the new version of the manuscript).

Reviewer 2 Report

The authors have conducted pilot retrospective analysis of a cohort of pediatric patients with acute appendicitis with the aim of detecting the relation with acute kidney injury. 

They have gathered data of all patients younger than 14 years of age, treated in their hospital during a six month period in 2020. They included patients who underwent appendectomy as well as those treated conservatively. The latter had positive ultrasound criteria for appendicitis (non compressible appendix having more than 6 mm in diameter) which is a sufficiently reliable tool for confirming but not for ruling out the diagnosis. 

It remains unclear what were the indications for non-operative treatment of specific subgroups of patients. It would be interesting to see how they were treated (observation only, IV fluids, antibiotics yes/no?), for how long and were there any complications or need for surgical intervention in the future. 

The idea behind the study is simple but original. I was surprised to learn that there were no studies documenting the relation between appendicitis and acute kidney injury.  

The introduction section is sufficient in presenting the background of the study. The authors clarify the importance of detecting the episodes of AKI in children which doubles the risk of chronic kidney disease. The hypothesis is clear and well presented. 

The design of the study could be improved, because prospective study could give much more information and produce stronger conclusions. The authors frankly give the drawbacks of their current study at the end of the Discussion section. I assume that since this study has been labeled as "pilot", it will serve as a leverage for designing and conducting an even broader prospective study which could collect more data. 

The authors refer to the Ethical Committee approval for the current study claiming that informed consent was obtained from all subjects, but surely they mean consent before surgery or before admittance to hospital, so I don't think it's relevant for retrospective anonymous data collection, otherwise it should be clarified what was approved by Ethical Committee. 

The manuscript is clear, it is relevant for everyday practice, and it is presented in a well-structured manner. The results are reproducible and the conclusions are sound and based on the study findings. 

I encourage the authors to design and conduct a protocol for detecting the AKI in children with AA, thus overcoming the study limitations appropriately identified in the manuscript. 

Author Response

Reviewer 2

The authors have conducted pilot retrospective analysis of a cohort of pediatric patients with acute appendicitis with the aim of detecting the relation with acute kidney injury. 

They have gathered data of all patients younger than 14 years of age, treated in their hospital during a six-month period in 2020. They included patients who underwent appendectomy as well as those treated conservatively. The latter had positive ultrasound criteria for appendicitis (non-compressible appendix having more than 6 mm in diameter) which is a sufficiently reliable tool for confirming but not for ruling out the diagnosis. 

It remains unclear what were the indications for non-operative treatment of specific subgroups of patients. It would be interesting to see how they were treated (observation only, IV fluids, antibiotics yes/no?), for how long and were there any complications or need for surgical intervention in the future. 

Answer: At admission, all the patients –independently from the chosen treatment modality– underwent intravenous hydration with Glucose 5% Sodium Chloride 0.9% with a mean duration of intravenous fluids was of 3.7days±0.7SDS. We also added more details about duration of intravenous fluids administration and we discussed these findings in discussion section. Moreover, the patients conservatively managed underwent treatment with intravenous cephalosporins and metronidazole for 3–5 days followed by oral amoxicillin and clavulanic acid for 10 days. As per our clinical practice the patients were conservatively managed in case the clinical findings were suggestive of non-complicated AA (appendicitis without neither perforation nor appendiceal abscess nor mass formation). Because often the differentiation between uncomplicated and complicated or perforated appendicitis may be uncertain before the operation(doi:10.1001/jamapediatrics.2017.0057), in case of uncertainty, the surgical treatments was chosen. We added this information in the new version of the manuscript (please see lines 117-119, and 126-137, 202-230, and Table 1 of the new version of the manuscript).

We did not collect data about possible complications or need for AA surgical intervention in the future post-discharge period because this is outside of the aims of the current study and in literature there are several studies addressing this interesting issue (doi:10.1001/jamapediatrics.2017.0057; doi: 10.1007/s00431-017-2867-2; doi:http://dx.doi.org/10.1016/j.ijscr.2015.05.003; doi: DOI:10.1097/MOP.0000000000000487; doi: 10.1542/peds.2016-3003).

The idea behind the study is simple but original. I was surprised to learn that there were no studies documenting the relation between appendicitis and acute kidney injury.  The introduction section is sufficient in presenting the background of the study. The authors clarify the importance of detecting the episodes of AKI in children which doubles the risk of chronic kidney disease. The hypothesis is clear and well presented. 

Answer: thank you.

The design of the study could be improved, because prospective study could give much more information and produce stronger conclusions. The authors frankly give the drawbacks of their current study at the end of the Discussion section. I assume that since this study has been labeled as "pilot", it will serve as a leverage for designing and conducting an even broader prospective study which could collect more data. 

Answer: yes, this study will open the way to a prospective study with a specific post-discharge nephrological follow-up in order to precisely evaluate the recovery from AKI and the onset of possible AKI complications such as reduced eGFR, proteinuria or hypertension during the follow-up. We added these considerations in the new version of the manuscript (please see lines 251-254).

The authors refer to the Ethical Committee approval for the current study claiming that informed consent was obtained from all subjects, but surely they mean consent before surgery or before admittance to hospital, so I don't think it's relevant for retrospective anonymous data collection, otherwise it should be clarified what was approved by Ethical Committee. 

Answer: yes, this consent is referred to the made clinical, biochemical, instrumental and surgical procedures. We clarified it (please see line 273 of the new version of the manuscript).

The manuscript is clear, it is relevant for everyday practice, and it is presented in a well-structured manner. The results are reproducible and the conclusions are sound and based on the study findings. I encourage the authors to design and conduct a protocol for detecting the AKI in children with AA, thus overcoming the study limitations appropriately identified in the manuscript. 

Answer: thank you. We have planned to design a prospective study.

Reviewer 3 Report

First, I would like to congratulate the authors for conducting this research. In this retrospective study, the authors included 122 children with clinico-radiological confirmation of appendicitis. The authors not only investigated the prevalence of AKI in pediatric AA but also explored the factors associated with AKI in their cohort.

It was observed that AKI existed in 7% of children with AA. Also, when compared with cases without AKI, children with AKI showed a higher prevalence of fever≥38.5°C (p=0.02), vomiting (p=0.03), and ≥5%dehydration (p=0.03), and higher levels of both C-reactive protein (CRP) (p=0.002) and neutrophils (p=0.03).

The study has merit and will be of interest to our readers. However, I have some comments that need to be addressed before the manuscript can be reconsidered.

Abstract: Please make a structured abstract and divide it into background, methods, results, and conclusion sections.

Introduction: This section is very brief. Please expand it.

-Please add 2-3 lines about your hypothesis at the end of this section. What did you hypothesize before conducting this study?

Methods: It is well-written. No major changes are needed.

-The authors have excluded 15 patients with COVID-19 infection and 20 cases with missing data about serum creatinine values at admission. This can be mentioned in the results section. The authors can maybe start the results section with a total n=157.

Results: Please check the association of the severity of appendicitis (complicated vs uncomplicated) with AKI. This variable has not been addressed.

-Also, I would advise the authors not to conduct a regression analysis in such a limited sample (n=9). There is no point in demonstrating Table 2. Please remove it.

-Please correct the grammatical errors by taking the help of a writing assistant.

Author Response

Reviewer 3

First, I would like to congratulate the authors for conducting this research. In this retrospective study, the authors included 122 children with clinico-radiological confirmation of appendicitis. The authors not only investigated the prevalence of AKI in pediatric AA but also explored the factors associated with AKI in their cohort. It was observed that AKI existed in 7% of children with AA. Also, when compared with cases without AKI, children with AKI showed a higher prevalence of fever≥38.5°C (p=0.02), vomiting (p=0.03), and ≥5%dehydration (p=0.03), and higher levels of both C-reactive protein (CRP) (p=0.002) and neutrophils (p=0.03). The study has merit and will be of interest to our readers. However, I have some comments that need to be addressed before the manuscript can be reconsidered.

Answer: thank you.

Abstract: Please make a structured abstract and divide it into background, methods, results, and conclusion sections.

Answer: according to the instructions for authors of Children, the abstract should be a single paragraph and should follow the style of structured abstracts, but without headings. For this reason, the abstract has the stile of structured abstracts with background, aims, methods, results, and conclusions but without headings.

Introduction: This section is very brief. Please expand it.

-Please add 2-3 lines about your hypothesis at the end of this section. What did you hypothesize before conducting this study?

Answer: we expanded introduction and we added more information about our hypotheses (please see lines 34-60 of the new version of the manuscript).

Methods: It is well-written. No major changes are needed.

Answer: thank you.

-The authors have excluded 15 patients with COVID-19 infection and 20 cases with missing data about serum creatinine values at admission. This can be mentioned in the results section. The authors can maybe start the results section with a total n=157.

Answer: we modified the results section accordingly (please see lines 113-116 of the new version of the manuscript).

Results: Please check the association of the severity of appendicitis (complicated vs uncomplicated) with AKI. This variable has not been addressed.

Answer: we added this information in the new version of the manuscript and we discussed it in the discussion section (please see lines 127-129 and 207-215 of the new version of the manuscript).

-Also, I would advise the authors not to conduct a regression analysis in such a limited sample (n=9). There is no point in demonstrating Table 2. Please remove it.

Answer: we prefer to not delete the Table 2 to give an idea of associations of these parameters with AKI. However, considering the limited sample size we better specified that this is only an exploratory analysis (please see lines 153-158 and 245-246 of the new version of the manuscript)..

-Please correct the grammatical errors by taking the help of a writing assistant

Answer: the written English was carefully revised.

Round 2

Reviewer 1 Report

All previous suggestions have been addressed. No further comments.

Reviewer 3 Report

The authors have incorporated all my comments in the revised manuscript. The overall scientific quality of the manuscript has improved significantly. I would like to thank the authors for submitting their work to MDPI Children.